# Evaluation of the Retentive Forces from Removable Partial Denture Clasps Manufactured by the Digital Method

Vitor Anes [1,2,3,*], Cristina B. Neves [4,*], Valeria Bostan [5], Sérgio B. Gonçalves [3] and Luís Reis [3]

1    Instituto Superior de Engenharia de Lisboa, 1959-007 Lisboa, Portugal
2    Instituto Politécnico de Lisboa, 1549-020 Lisboa, Portugal
3    IDMEC, Instituto Superior Técnico, Universidade de Lisboa, 1049-001 Lisboa, Portugal;
     sergio.goncalves@tecnico.ulisboa.pt (S.B.G.); luis.g.reis@tecnico.ulisboa.pt (L.R.)
4    Biomedical and Oral Sciences Research Unit (UICOB), Faculdade de Medicina Dentária,
     Universidade de Lisboa, 1600-277 Lisboa, Portugal
5    Faculdade de Medicina Dentária, Universidade de Lisboa, 1600-277 Lisboa, Portugal;
     valeriabostan@campus.ul.pt
*    Correspondence: vitor.anes@isel.pt (V.A.); mneves@edu.ulisboa.pt (C.B.N.)

**Abstract:** The purpose of this study was to evaluate the retentive forces over time of removable partial denture clasps fabricated by the digital method. Occlusal rest seats were fabricated on three premolar teeth fixed in acrylic blocks ($9 \times 20 \times 40$ mm). Digitization of the teeth was performed using a laboratory scanner (Zirkonzahn Scanner S600 GmbH, Gais, Italy). After the analysis and determination of the insertion axis, two types of clasps with mesial occlusal rests were designed per tooth: the back-action and the reverse back-action clasps, using the Partial Planner Zirkonzahn program. The file was sent for fabrication of six metal clasps from a cobalt-chromium SP2 alloy in the EOSINT M270 system by a direct laser sintering process. The Instron 5544 universal testing machine was used to perform 20,000 cycles of clasp insertion and removal in the corresponding tooth with a load cell of 100 N and a speed of 2.5 mm/s. The retentive force was recorded for each of the 1000 cycles, and the change in retention over time was calculated. Statistical analysis was performed using the nonparametric Mann–Whitney test and a significance level of 5%. At 16,000 cycles, a maximum change in retention of 3.74 N was recorded for the back-action clasps and a minimum of $-24.28$ N at 1000 cycles for the reverse back-action clasps. The reverse back-action clasps exhibited statistically significant lower change in retention than the reverse-action clasps at 4000 and 5000 cycles. No differences were observed in the remaining cycles. During the 20,000 cycles, the change in retention was low regardless of the type of clasp. For most cycles, there were no differences in the change in retention between the two types of clasps.

**Keywords:** removable partial denture metal framework; direct laser sintering; back-action clasp; reverse back-action clasp; retentive force; experiments; additive manufacturing; CAD-CAM





## 1. Introduction

Over the years and with scientific and technological progress, society has given more importance to oral health maintenance, promoting campaigns and prevention programs while developing dental materials and treatment options [1–3]. These efforts have led to a decrease in tooth loss during life and an increase in the number of cases of partial rather than complete edentulism [1–4].

In addition, mortality has decreased, and average life expectancy has increased, leading to an aging society [2,3,5]. For this reason, the need for treatment with fixed or removable prostheses is increasing every year [2]. In this sense, the rehabilitation of edentulous patients is crucial for their quality of life as it preserves masticatory function, phonetics and esthetics [1,6].

Fixed rehabilitation with implants has high success rates and has become increasingly popular in recent years. However, not all patients are good candidates for this type of treatment due to health, financial, anatomical, or psychological circumstances [7]. One alternative is the rehabilitation of edentulous areas with removable prostheses, which is still widely used in clinical practice [5,8]. Removable partial dentures (RPD) are prostheses that restore edentulous areas in partial edentulism, and can be performed rapidly with esthetic and functional benefits [5,9–11].

The development of this type of prosthesis remains a challenge. Proper planning and selection of RPD materials is key to reducing the incidence of complications.

When designing RPD, it is important that the framework has stability and a correct fit which is mostly provided by the major connector and clasps [12]. It should also provide support and retention, and not interfere with abutment teeth and other supporting structures when masticatory forces are applied [13,14]. The choice of the correct clasp geometry starts with the analysis of the Kennedy class and the insertion axis of the prosthesis, which determines the equator and the retentive zone of the tooth [12,13]. A clasp assembly consists of an occlusal rest, a reciprocal arm and a retentive arm [12]. All clasps must provide retention, stability, support, reciprocity and passivity and cover the largest possible area of the tooth [12,15,16]. The retentive arm provides primary retention of the prosthesis and must be able to deform as it passes the equator of the tooth and return to its original position without exerting damaging forces on the tooth [15,17,18].

In partial edentulous patients with bilateral (Class I Kennedy) or unilateral (Class II Kennedy) distal extension, retention, stability and support are very important [19]. The selected clasp must provide good retention, transfer forces parallel to the tooth axis, and minimize the application of damaging forces to the abutment tooth [12,19]. In these situations, it is common to use clasps with mesial occlusal rests, such as the reverse back-action clasp when buccal retention is on the distal side and the back-action clasp when there is mesial retention [16,19,20].

Of the various materials available for the manufacture of the metal framework, cobalt–chromium (Co–Cr) alloy has been the most commonly used material since 1929 [21]. This material is known for its low cost and adequate mechanical properties, such as lower density and higher modulus of elasticity than gold, as well as its high corrosion resistance, which is related to biocompatibility [4,17,18,21]. This metallic alloy contains about 53% to 67% Co, 25% to 32% Cr, 2% to 6% Mo, and a mixture of other elements [22,23].

The lost waxing casting or conventional method is most commonly used to produce metal frameworks [9,23]. The quality of the production largely depends on the experience of the laboratory technician and the quality of the impressions taken by the clinician, which is a time-consuming and expensive method [9,24]. Normally, this technique is prone to several errors that can result in 75% of frameworks not fitting properly on the supporting structures at the time of insertion into the patient's oral cavity [12,25,26]. Thus, over the years, the need to develop new techniques has increased, and scientific advances have made it possible to improve the process of RPD framework production, reducing the time and cost of fabrication, and improving the fit and functional efficiency of rehabilitations [1,3].

With the advent of digital technology, manufacturing methods have changed with the introduction of digital processes such as computer-aided design (CAD) and computer-aided manufacturing (CAM) [6,27]. With the development of these new methods, the results have become more predictable, reproducible, and accurate, which improves the longevity of the rehabilitations [4,28]. The use of CAD/CAM offers the advantages of digital planning and analysis, reduction in material waste and use of innovative materials, better communication between laboratory and clinician, reduction in the number of steps and errors, easy cast reproduction, and better quality control of the production [4,29–31].

The production of these structures can be performed by two digital methods: addition (AM—additive manufacturing) or subtraction (milling) [29]. The AM method makes use of sintering technology, such as: SLM—Selective Laser Melting, SLS—Selective Laser Sintering or DMLS—Direct Metal Laser Sintering [9,18]. This technology offers high productivity,

improves the properties of materials by increasing density and homogeneity, and is a precise and cost-effective method [4,9]. Basically, additive manufacturing is based on the production of metal objects in 3D by sintering metal powders with a high-power laser [9,32]. In DMLS, a powder consisting of several metals is partially fused with a high-power Yb fiber laser [18,32]. This technique uses a powder with a mixture of metals whose melting temperatures are different to produce the solid metal structure [9,18]. The composition of Co–Cr powder is mainly Co and Cr, but also contains metals such as Mo, W, Si, Ce, F, Mn and C [33,34].

Since there is little information on the behavior of metal framework production by the digital method, it is important to evaluate the change over time of the retention forces of the RPD clasps.

In this sense, the aim of this study was to evaluate the retentive forces and the change in retention over time of RPD clasps produced by the digital method. Another objective was to compare the retentive force of clasps fabricated with different designs.

## 2. Materials and Methods

### 2.1. Production of Models

Three different intact premolar teeth were selected from a reservoir of the BIOMAT laboratory of the Faculty of Dental Medicine, University of Lisbon. Longitudinal undercuts were created around each tooth root to increase its adherence to the resin block. Then, it was scanned with the Zirkonzahn Scanner S600 equipment (Zirkonzahn GmbH, Gais, Italy) (Figure 1).

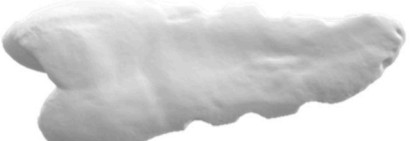

**Figure 1.** Digitized premolar with undercuts.

In the Autodesk Meshmixer program (2017, version 3.5.474), a 3D-printed resin block measuring $9 \times 20 \times 40$ mm was designed for each tooth so that the long axis of the sinus (negative) of the root was perpendicular to the occlusal plane. The block was then printed using the NextDent 5100 3D printer (NextDent BV, Soesterberg, The Netherlands) (Figure 2), using NextDent Model 2.0 resin (Next Dent BV, Soesterberg, The Netherlands). Each tooth was embedded in one acrylic resin block and placed in a LC-3DPrint Box unit (NextDent, BV, Soesterberg, The Netherlands) for light curing for 30 min (Figure 3).

Mesial occlusal rest seats were performed on the selected teeth (Figure 4), according to the support principles: rounded triangular and concave shape, angle formed by the occlusal rest and the minor connector lower than 90°, minimum thickness of 1 mm and extension of one third in the mesiodistal and buccolingual lengths [12].

Finally, the model was scanned with a laboratory scanner (S600 Arti, Zirkonzahn GmbH, Italy) (Figure 5) and the final digital model was created, on which the clasps were later designed.

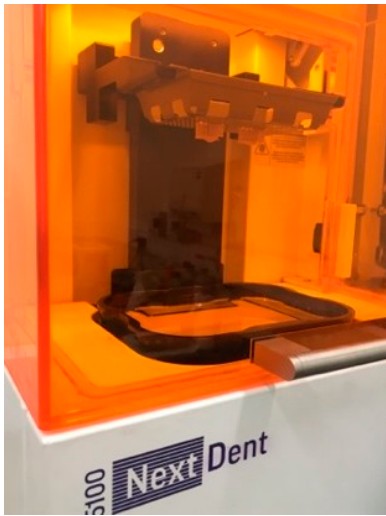

**Figure 2.** Next Dent 5100 3D printing machine, 3D Systems.

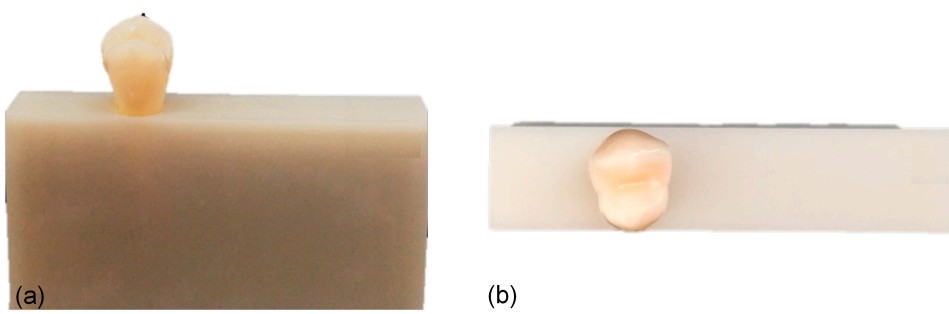

(a)　　　　　　　　　　　　　　　　　　　　　(b)

**Figure 3.** Acrylic block with embedded tooth 1: (**a**) Lingual view, (**b**) Occlusal view.

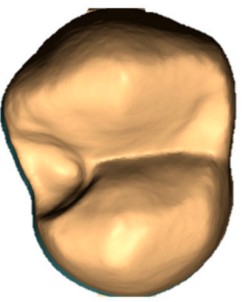

**Figure 4.** Occlusal view of mesial occlusal rest seat of tooth 1.

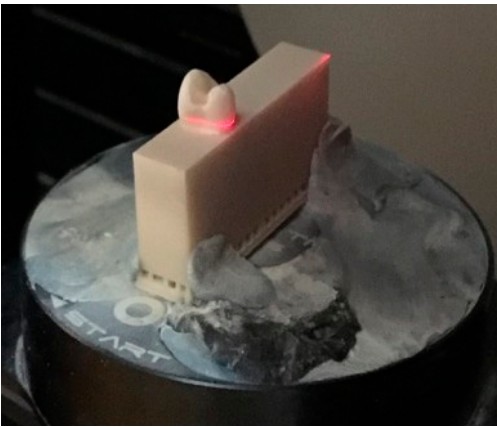

**Figure 5.** Digitization of the tooth with occlusal rest seat.

### 2.2. Production of Clasps

The digitized model was used with the clasp design using the Partial Planner program (Zirkonzahn, GmbH, Italy). Two types of clasps were designed with mesial occlusal rest, reciprocal arm in lingual tooth surface and retentive arm in the buccal tooth surface: back-action clasp (active tip of retentive arm in mesial direction) and reverse back-action (active tip of retentive arm in distal direction). In both cases, the longitudinal axis of the tooth was determined and used as the insertion axis for the clasp. The equator of the tooth was determined, and the retentions were removed. Then, in the case of the back-action clasp, a retention of 0.25 mm was sought on the mesial buccal surface, while in the case of the reverse back-action clasp, a retention of 0.25 mm was sought on the distal buccal surface.

After this step, the position of the retentive and reciprocal arms of the two clasps were determined. The reciprocal arm and the body of the retentive arm were located above the equator of the tooth, while the tip of the retentive arm was located below, in the previously selected zone. In addition, the mesial occlusal rest and the distal minor connector were added (Figures 6 and 7) [12]. A cylinder with a diameter of 5 mm and a height of 20 mm was attached to the distal end of the minor connector to serve as a support for the test machine (Figure 8). Finally, a digital design of each clasp in standard tessellation language (STL) file was created and sent to a commercial laboratory production center (Sineldent, Spain).

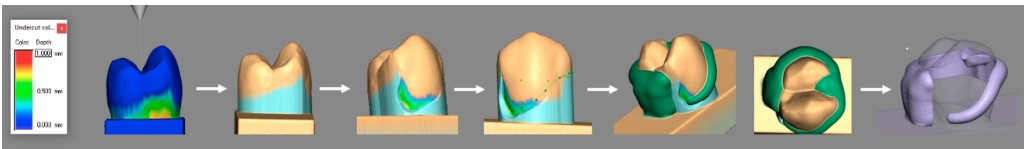

**Figure 6.** Diagram of the steps for making the back-action clasp.

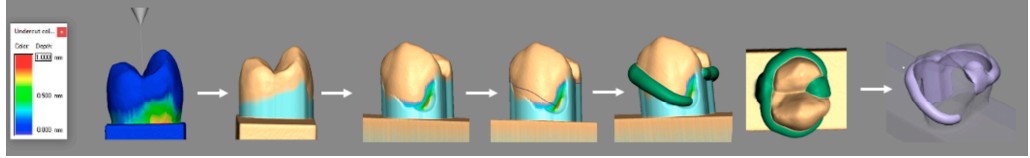

**Figure 7.** Diagram of the steps for making the reverse back-action clasp.

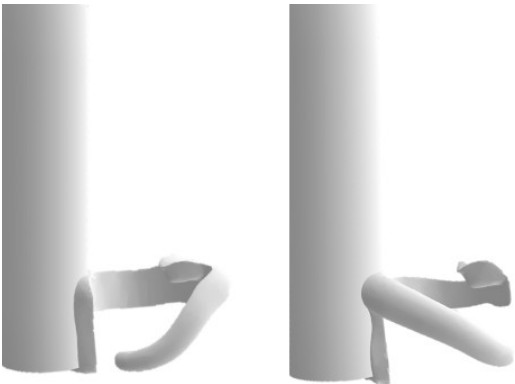

**Figure 8.** Clasps, final testing model.

A total of six Co–Cr clasps were fabricated, namely, two back-action clasps and two reversal back-action clasps for each of the three selected teeth.

The clasps were fabricated from SP2 Co–Cr alloy (EOS GmbH, Krailling, Germany) with 420 HV of hardness and 1350 MPa of tensile strength [35], using an EOSINT M270 system (EOS GmbH, Germany) with the direct metal laser sintering method and then heat treated for 45 min to remove the internal stresses of the metal. The clasps were then

placed in an electrolytic bath (Polytherm compact, Dentaurum GmbH & Co. KG, Ispringen, Germany) for 3 min and finished with brushes, special polishing rubbers, and polishing paste by the same technician to diminish the high roughness of the metal surface. After this step, the fit of the clasps on the tooth was tested. A good fit was considered to be when the occlusal rest rested on the respective seat, in continuity with the tooth and when the clasp arms were in contact with the respective tooth surface (Figure 9).

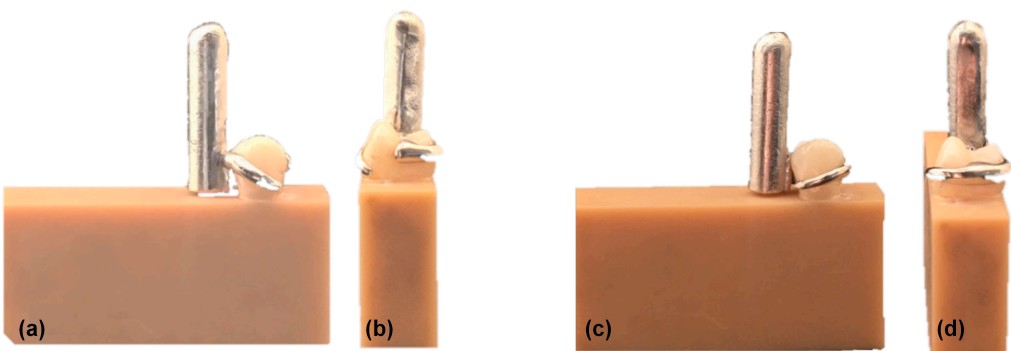

**Figure 9.** Back-action clasp, of tooth 2: (**a**) buccal view, (**b**) mesial view and reverse back-action clasp of tooth 2 (**c**) buccal view, (**d**) mesial view.

### 2.3. Test Conditions

For the evaluation of the retentive forces, repetitive cycles of insertion and removal of the clasps were performed using a universal mechanical testing machine Instron 5544 Tensile Tester (Instron, Norwood, MA, USA) equipped with a 100 N load cell. Mechanical claws were used to fix the resin block in which the tooth is fixed (BioPlus, Cat.: 2752-005, Charlotte, NC, USA) and a drill adapter with serial number 107,943 (Instron, Norwood, MA, USA) was used to fix the vertical cylinder to the clasp (Figures 10 and 11).

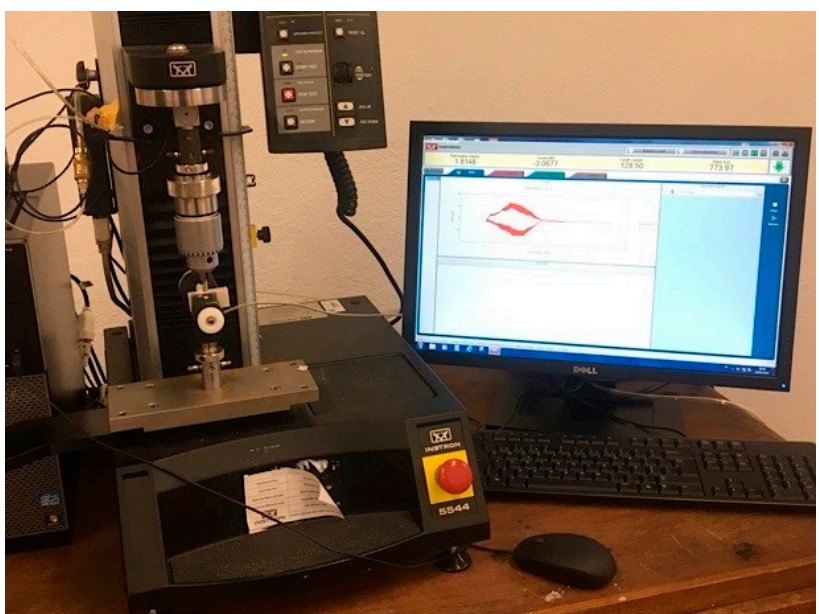

**Figure 10.** Instron 5544 tensile mechanical testing machine.

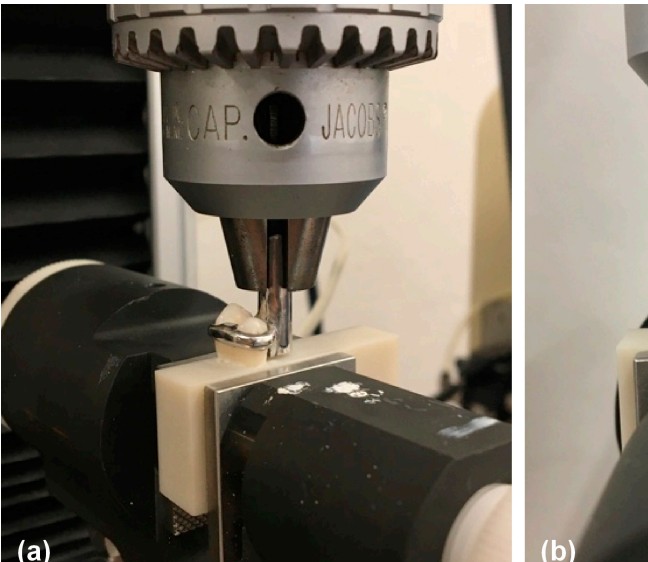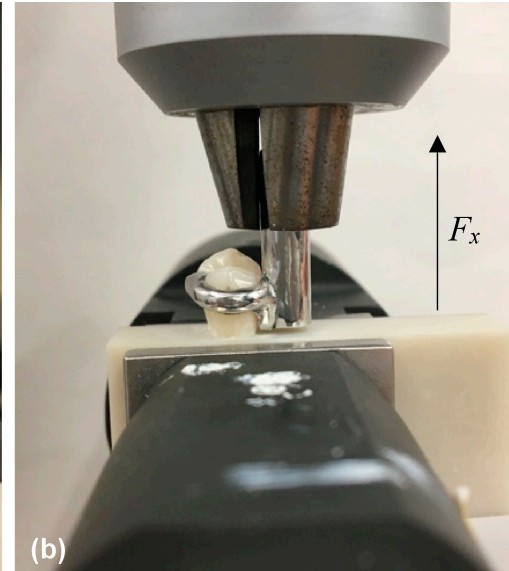

**Figure 11.** Mounting of the test models on the test device, reverse back-action clasp on tooth 1: (**a**) perspective view, (**b**) lingual view.

A total of 20,000 cycles of vertical movements, parallel to the longest axis of the tooth and to the axis of insertion of the clasps, were performed to simulate thirteen years of usage of the RPD, after assuming that the patient performs four cycles of insertion and removal of the RPD per day. This movement was performed at a constant rate of 2.5 mm/s. The force required to remove each clasp was recorded using the Bluehill version 3.0 program (Instron, Norwood, MA, USA).

The change in retention when the clasp is removed was calculated using Equation (1):

$$\Delta F = F_0 - F_x \tag{1}$$

In the formula, $\Delta F$ corresponds to the change in retention, $F_0$ to the force required to remove the clasp at 0 cycles, and $F_x$ to the force required to remove the clasp after $x$ cycles. The retentive forces were recorded every 1000 cycles until 20,000 cycles were reached. The percentage change in retention was also calculated using Equation (2):

$$\frac{\Delta F}{F_0} \cdot 100\% \tag{2}$$

For the teeth tested, the buccal and lingual surfaces were observed before and after the tests using an optical microscope (Nikon Optiphot, Tokyo, Japan) and a stereo zoom microscope (Optika Microscopes. SLX series, Ponteranica, Italy). On each tooth, points were marked on the buccal surface—mesial and distal—and on the lingual surface of the tooth for reference. A photographic evaluation of the wear of the tooth was made using the photographic record before and after the tests for each clasp.

### 2.4. Statistical Analysis

A descriptive analysis of the retentive forces of each clasp was performed every 1000 cycles and for each tooth. The median and interquartile range of change in retention and the percent change in retention by clasp type were calculated. Inferential statistical analysis of change in retention by clasp type was performed using the nonparametric Mann–Whitney test. The significance level was set at 5% ($\alpha = 0.05$).

### 3. Results

Figure 12 shows the retentive forces of the two types of clasps for each of the three teeth used. The back-action clasp had an initial retentive force (cycle = 0) of 9.64 N, 10.87 N,

and 11.24 N on tooth 1, 2, and 3, respectively, and a final retentive force (cycle = 20,000) of 8.88 N, 11.78 N, and 9.55 N, respectively. The reverse back-action clasp had an initial retentive force (cycle = 0) of 8.58 N, 12.97 N, and 11.76 N on tooth 1, 2, and 3, respectively, and a final retentive force (cycle = 20,000) of 12.10 N, 10.49 N, and 12.05 N, respectively.

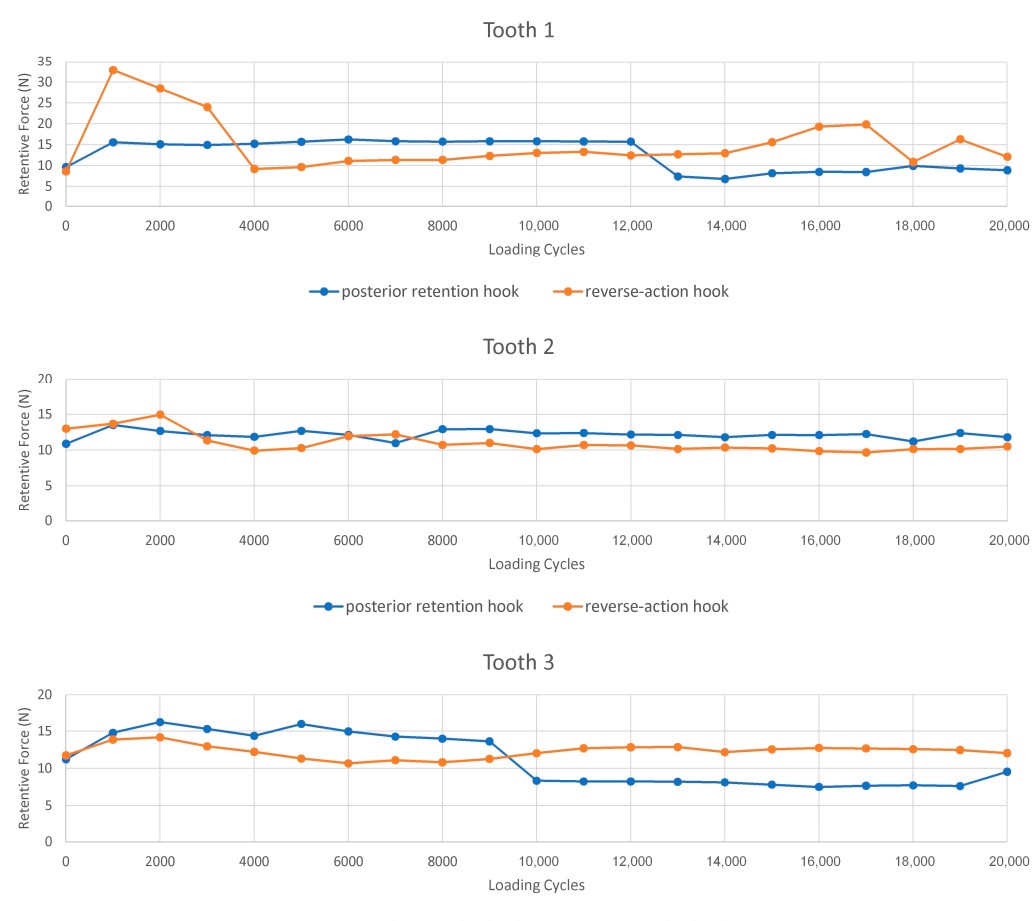

**Figure 12.** Diagram showing the retentive forces of the two clasps over the cycles on tooth 1, 2 and 3.

In tooth 1, the reverse back-action clasp initially showed higher values for retentive force than those of the back-action clasp. At 4000 cycles these values decreased, and at 13,000 cycles exceeded the values of the back-action clasp (Figure 12). In tooth 2, the reverse back-action clasp showed higher values of retentive force at the beginning of the test compared to the back-action clasp, but at 8000 cycles it had lower values. The back-action clasp initially had higher values of retentive forces than the reverse back-action clasp, but at 10,000 cycles it showed lower values (Figure 12).

Figures 13 and 14 show the percent change in retention over cycles by clasp type. For teeth 1 and 3, the back-action clasps experienced an abrupt loss of retention at 9000 and 13,000 cycles, respectively (Figure 13).

In the reverse back-action clasps, there were a few losses of retention over time in teeth 2 and 3 and an apparent loss of retention up to 4000, followed by a slight increase in retention in tooth 1 (Figure 14).

Descriptive analysis of the change in retention over the 20,000 cycles by clasp type is shown in Figure 15 and Table 1. The table shows the median, interquartile range, and maximum and minimum values of the variation in retention for each clasp type. There was a maximum force variation of 3.74 N (16,000 cycles for the reverse back-action clasp) and a minimum of −24.28 N (1000 cycles for the reverse back—action clasp). Negative values of change mean that the retentive force increases with the number of cycles and positive values of change mean that the retentive force decreases.

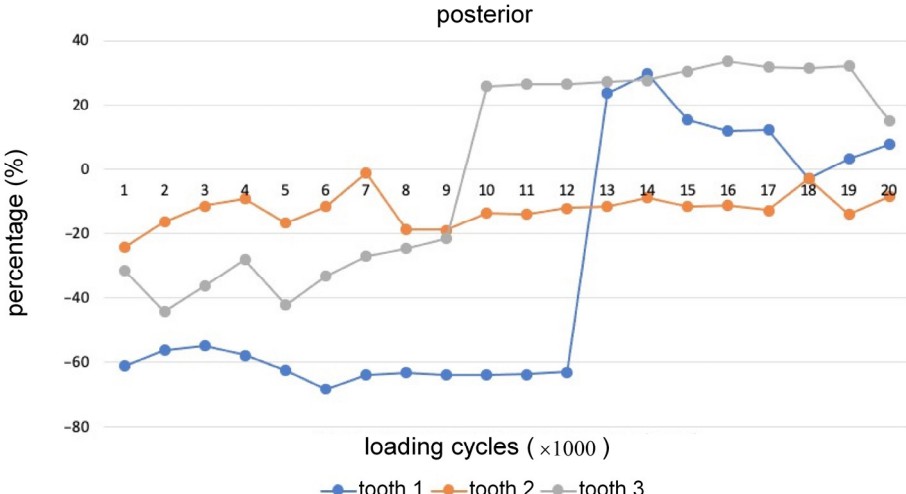

**Figure 13.** Percentage change in retention over loading cycles—back-action clasp on each tooth.

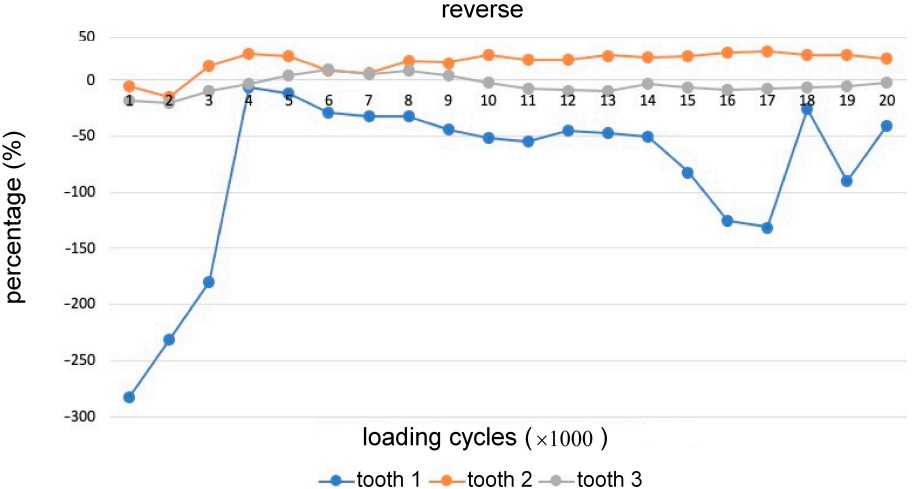

**Figure 14.** Percent change in retention over loading cycles—reverse back-action clasp on each tooth.

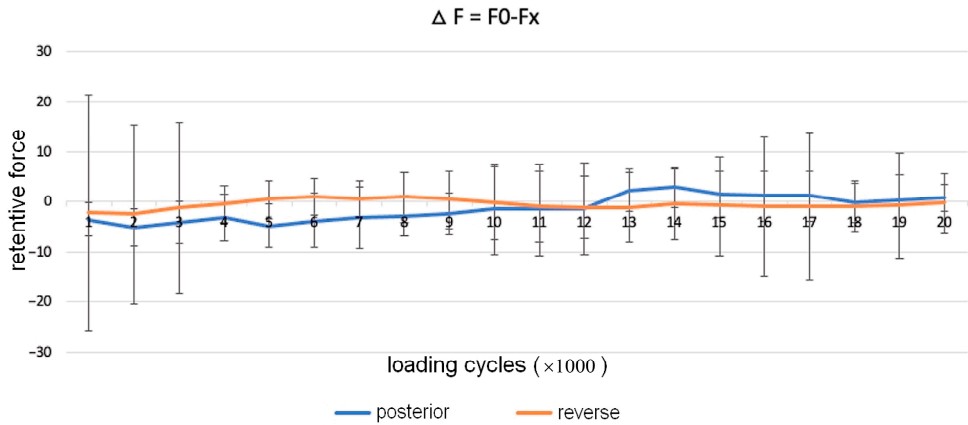

**Figure 15.** Change in retention of the two types of claps during insertion and removal.

**Table 1.** Median, interquartile range, maximum and minimum variation of retentive forces ($\Delta F = F_0 - F_x$) by clasp type, units in [N].

| Cycles | Back-Action Clasp | | | | Reverse Back-Action Clasp | | | |
|---|---|---|---|---|---|---|---|---|
| | Median | Amplitude Interquartile | Maximum | Mínimum | Median | Amplitude Interquartile | Maximum | Minimum |
| It 1000 | −3.53 | 3.263 | −2.64 | −5.90 | −2.12 | 23.587 | −0.69 | −24.28 |
| 2000 | −4.98 | 3.655 | −1.77 | −5.43 | −2.42 | 17.883 | −1.97 | −19.85 |
| 3000 | −4.07 | 4.062 | −1.22 | −5.28 | −1.21 | 17.044 | 1.62 | −15.41 |
| 4000 | −3.14 | 4.611 | −0.97 | −5.58 | −0.46 | 3.627 | 3.03 | −0.58 |
| 5000 | −4.73 | 4.221 | −1.80 | −6.03 | 0.44 | 3.738 | 2.68 | −1.05 |
| 6000 | −3.72 | 5.36 | −1.23 | −6.59 | 1.03 | 3.623 | 1.09 | −2.53 |
| 7000 | −3.04 | 6.050 | −0.10 | −6.16 | 0.65 | 3.576 | 0.77 | −2.79 |
| 8000 | −2.75 | 4.079 | −2.02 | −6.10 | 0.93 | 5.045 | 2.24 | −2.80 |
| 9000 | −2.40 | 4.115 | −2.05 | −6.16 | 0.48 | 5.730 | 1.98 | −3.74 |
| 10,000 | −1.46 | 9.067 | 2.89 | −6.16 | −0.27 | 7.272 | 2.83 | −4.43 |
| 11,000 | −1.50 | 9.133 | 2.98 | −6.14 | −0.93 | 6.987 | 2.27 | −4.71 |
| 12,000 | −1.29 | 9.069 | 2.98 | −6.08 | −1.09 | 6.183 | 2.32 | −3.86 |
| 13,000 | 2.27 | 4.284 | 3.04 | −1.23 | −1.11 | 6.923 | 2.82 | −4.10 |
| 14,000 | 2.84 | 4.053 | 3.11 | −0.93 | −0.42 | 6.993 | 2.62 | −4.36 |
| 15,000 | 1.49 | 4.658 | 3.42 | −1.23 | −0.79 | 9.792 | 2.75 | −7.04 |
| 16,000 | 1.16 | 4.955 | 3.74 | −1.21 | −0.98 | 13.844 | 3.11 | −10.73 |
| 17,000 | 1.18 | 4.939 | 3.56 | −1.37 | −0.92 | 14.591 | 3.31 | −11.27 |
| 18,000 | −0.27 | 3.850 | 3.52 | −0.32 | −0.82 | 5.109 | 2.85 | −2.25 |
| 19,000 | 0.32 | 5.120 | 3.61 | −1.50 | −0.70 | 10.531 | 2.82 | −7.70 |
| 20,000 | 0.76 | 2.602 | 1.68 | −0.91 | −0.29 | 6.006 | 2.48 | −3.52 |

According to the inferential analysis, there was only a statistically significant difference between the change in retention of the two types of clasps at 4000 (Figure 16) and 5000 cycles (Figure 17). The reverse back-action clasp showed less change in retention at 4000 cycles ($p = 0.049$) and at 5000 cycles ($p = 0.049$) compared to the back-action clasp.

There were no statistically significant differences when comparing the change in retention in the remaining cycles between the two clasp groups ($p > 0.05$).

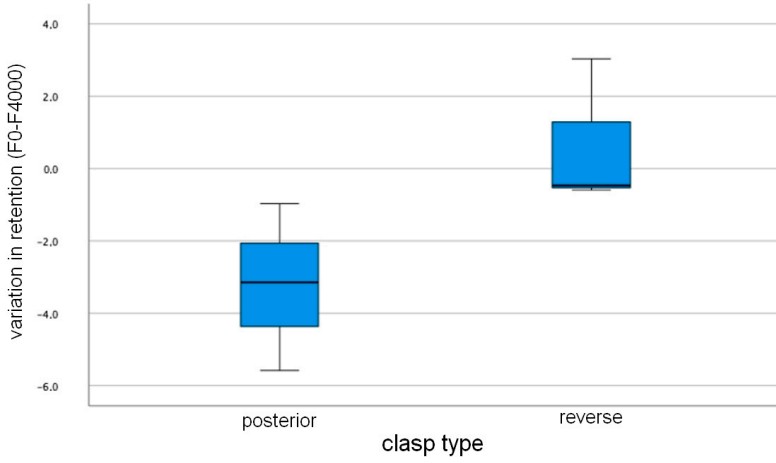

**Figure 16.** Boxplot diagrams of the change in retention of the two clasp types at 4000 cycles. Variation of retention in (N).

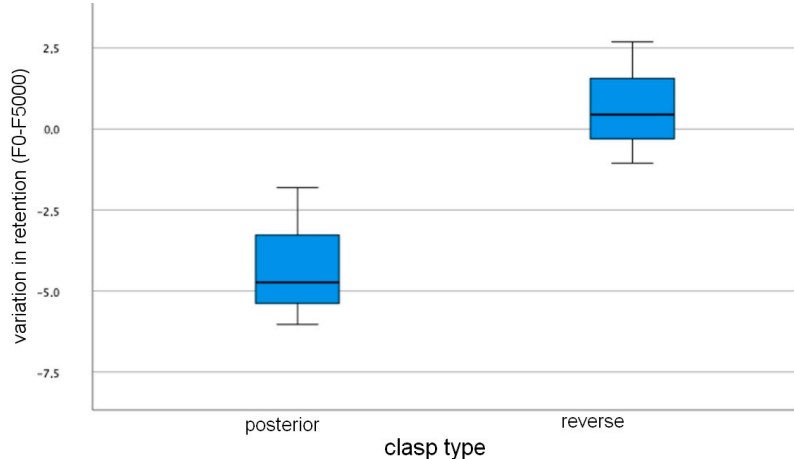

**Figure 17.** Boxplot diagrams of the change in retention of the two clasp types at 5000 cycles. Change in retention in (N).

## 4. Discussion

The objective of this study was to evaluate the retentive forces and their change over time of clasps with different designs that were digitally manufactured.

The prosthetic clasps are subject to daily movement during insertion and removal of the RPD by the patient. Several studies have examined the retentive force over multiple cycles of insertion and removal: 15,000 [8], 16,000 [15], 25,000 [34] and others [4,16,18,21,36,37]. In the present study, 20,000 cycles were performed, corresponding to a lifetime of 13 years, since changes in retention were observed only around this number of cycles.

The clasp retentive force is the force required to remove it from the tooth. It has been suggested that the retentive force for proper denture function is 5 N [12,21]. In the present study, it was shown that the initial retentive force of the back-action clasp was 10.58 N on average, and for the reverse back-action clasp was 11.10 N. This is consistent with the study of Tanaka, whose initial retentive force of Co–Cr clasps was $13 \pm 4$ N [38].

When evaluating the initial retentive forces versus the final retentive forces per tooth, the back-action clasp had lower initial and final retentive forces at tooth 1 compared to the reverse back-action clasp, but at tooth 2, the final retentive forces were higher for back-action forces. At tooth 3, the initial force of the back-action clasp was higher than that of the reverse back-action clasp, but the final force was lower. These discrepancies may be caused by differences in the tooth crown morphology. To address these discrepancies, the change in retention was calculated using Equation (1) to determine the variations in retention forces across cycles and to account for the initial retention force of each clasp. It is also important to highlight the fact that of the three teeth used, tooth 2 had a less retentive morphology and therefore had more constant values across cycles.

Both types of clasps have been shown to have some loss of retention over the loading cycles. However, the clasp with the back-action clasp was the one that showed the greatest change in retention at the end of the tests.

At the beginning of the experiments, both clasps showed a negative change in force, i.e., the retention values increased compared to the initial evaluation. This phenomenon was also found in other studies [4,8,36,39]. This can be justified because the tests were performed under cold conditions, and the insertion axis given by the testing machine was parallel to the longitudinal axis of the tooth, which is not the only axis that patients use when removing the removable framework [36]. The initial increase in retention forces can also be explained by the wear of the tooth and the inner surface of the clasp, which could have led to an increase in roughness at the beginning of the tests [8].

In Figures 13 and 14 the back-action clasp shows a tendency to decrease retention, with discrepancies between the values of the individual teeth. The reverse back-action clasp generally tends to maintain the values relatively constant. The fact that the reverse

back-action clasp has a longer arm may mean that it is more flexible, which may result in more constant values, while the back-action clasp with its shorter retention arm loses more retention over time [15].

In a study by Helal et al. the comparison of the retentive forces of two types of clasps, namely the back-action clap and the conventional Akers clasp, showed a statistically significant difference in retentive forces at 4000 cycles [15]. This is consistent with the results of the present study, which showed that design affected the change in retention at 4000 cycles ($p = 0.049$) and 5000 cycles ($p = 0.049$). However, there was no effect on the change in retention in the remaining cycles, which contradicts the results of Kato et al., Tannous et al. and Torii et al. 2018, that showed a continuous decrease in retentive forces during the test [8,17,34].

In agreement with the studies of Hebel et al. and Helal et al., tooth wear was observed [15,39] in the area where the active tip of the retention arm of the clasp contacts the tooth surface (Figures 18 and 19). This analysis was only a qualitative analysis, unlike the previously mentioned studies that used mathematical calculations to quantify wear. In this sense, wear was observed in all teeth and in both types of clasps on both the buccal and lingual sides of the tooth. This suggests that the clasps, although made with a digital method, also cause enamel wear.

However, as in previous studies, tooth wear was found to be very low, in the order of 20 microns, and therefore it can be assumed that the decrease in retentive force is not due to tooth wear but due to changes in the clasp [38,40]. In addition, it can be assumed that the retentive arm of the clasp causes minimal wear on the tooth surface over the years and regardless of the design.

The present study has several limitations, such as the fact that the three teeth used are morphologically different, which resulted in different retentive forces for the same type of clasp. This limitation was reduced by using variation of change values for the retentive force, which removed the influence of the initial retentive force of each clasp. Another limitation was that the tests were performed in a dry environment, which may result in higher frictional resistance between surfaces [36], and the result could have been different if the tests had been performed in an artificial saliva environment.

With the present study, it was possible to perform a comparison of the retentive forces and the change in retention over 13 years of use simulation between two different designs of clasps manufactured by the digital method.

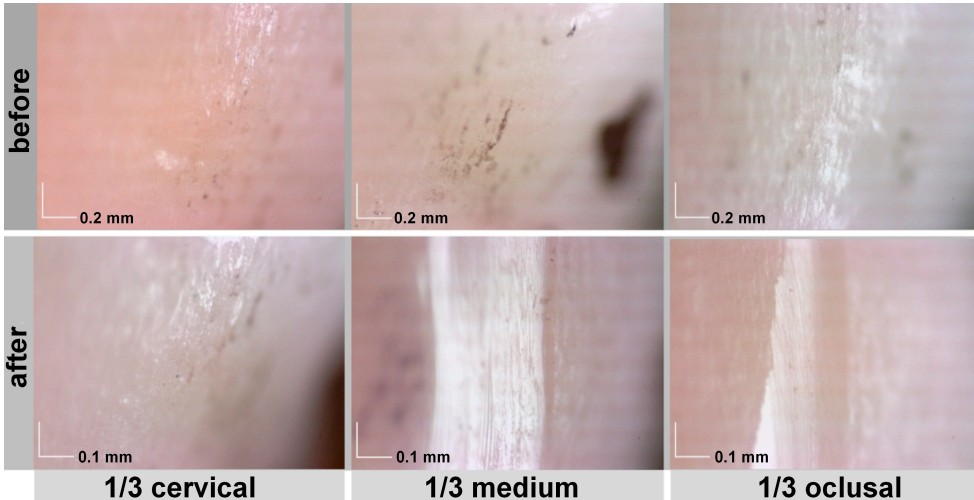

**Figure 18.** Buccal surface of tooth 3, before and after two trials with the back-action clasp, $5\times$ magnification, scale of the upper images of 0.2 mm and the lower of 1.0 mm.

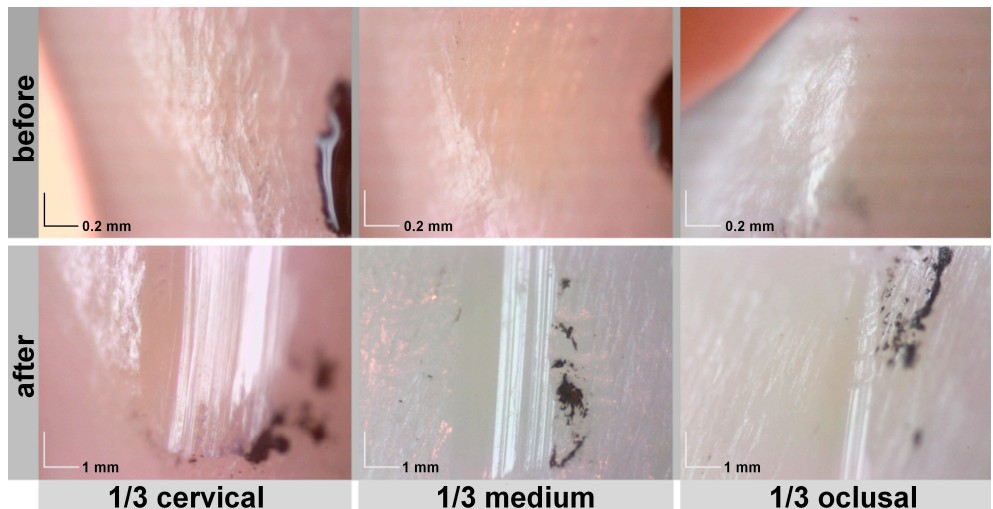

**Figure 19.** Distal and buccal surface of tooth 3, before and after two trials with the reverse back-action Clasp, 5× magnification, scale of the upper images is 0.2 mm and the lower is 1.0 mm.

The results show that, in general, there is no difference in retentive forces between the two types of clasps. However, there was no significant loss of retention force over time for either type of clasp, even when considering wear of the teeth, which, if relevant, would result in a loss of retentive force of the clasp.

In the future, it would be important to perform a comparison between the conventional and digital methods of RPD fabrication to determine if there are differences between these two clasp types, as there is little literature to compare them.

## 5. Conclusions

Based on the objectives of the present study and considering the limitations pointed out, it can be concluded that:

- Over 20,000 cycles, a reduced change in retention was verified in the clasps produced by the digital method, regardless of the type of clasp studied, which means that it will lose little retention over time.
- From this study, it can also be concluded that for most of the load cycles studied, no difference is observed between the changes in retention for the two types of clasps, leading to the conclusion that the design of the clasp does not have a great influence on the retentive force.

**Author Contributions:** Conceptualization, C.B.N., V.A. and V.B.; methodology, C.B.N.; software, S.B.G.; validation, V.A., C.B.N. and L.R.; formal analysis, C.B.N.; investigation, V.B.; resources, C.B.N. and L.R.; data curation, V.A.; writing—original draft preparation, V.B.; writing—review and editing, V.A.; visualization, V.B.; supervision, C.B.N.; project administration, C.B.N.; funding acquisition, C.B.N. All authors have read and agreed to the published version of the manuscript.

**Funding:** This research received no external funding.

**Institutional Review Board Statement:** Not applicable.

**Informed Consent Statement:** Not applicable.

**Acknowledgments:** This work was supported by FCT, through IDMEC, under LAETA, project UIDB/50022/2020.

**Conflicts of Interest:** The authors declare no conflict of interest.

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
