# Peer review of "Evaluation of the Retentive Forces from Removable Partial Denture Clasps Manufactured by the Digital Method"

_applsci, doi:10.3390/app13148072_

Round 1
Reviewer 1 Report
It would have been wonderful to see a comparison of the retentive properties of back action, reverse back action versus RPI and RPA designs in function of cyclic loading.
Author Response
The back-action and reverse back-action clasps are frequently used in our removable partial dentures designs since they allow assembly of minor connectors to come distally, not crossing the marginal gingiva from the supporting tooth, which is beneficial to the basic biologic principles of removable partial dentures. Nevertheless, the RPI and RPA clasps are typically designed for Class I and II distal extensions. They are described as the most effective clasps in diminishing the torque forces that can be transmitted to the supporting tooth. The authors appreciate this suggestion and will include it in further studies.
Reviewer 2 Report
The introduction section is wordy. Some paragraphs ought to be omitted.
There are a few mistakes that could be corrected
Author Response
The authors thank the reviewer for this suggestion. The introduction was revised and condensed with more explicit transitions between the themes.
Reviewer 3 Report
<General>
The review entitled “Evaluation of the retentive forces from removable partial dentures clasps manufactured by the digital method” has the aim to evaluate the retentive forces and the change in retention over time of RPD clasps produced by the digital method. The manuscript contains the new results. However, minor revisions are required to improve the paper.
My minor comments are as follows.
1) I wonder why you posted to Applied sciences. Is this content more suitable for the readers of Materials or Dentistry? From the conclusion, it seems that Materials is appropriate.
2) Page 2, Line 85-87
The quality of the production largely depends on the experience of the laboratory technician and the quality of the impressions taken by the clinician, which is a time-consuming and expensive method [9,24].
>It would be nice to have a concrete number to show how much less error there is in 3D scanning than the sum of traditional errors.
3) Page 4, Figure 3-5
What is the difference between the acrylic resins in the Figure3-5? Please tell us the characteristics of the three types and the reason for separating the resins. What are the effects of different resins?
4) Page 12, Line 343
These discrepancies may be because the three teeth used for this study have different clinical crown morphology.
> Is it common to see high reverse action values from 0-4000 cycles in Figure 14? What kind of anatomical difference is due to the fact that only tooth 1 behaves differently?
5) Page 13, Figure 22
The type of microscope is written in the text, and there is little point in including a photograph.
6) Page 13, Line 376-381
This part should move to Materials and Methods.
Author Response
Q1. I wonder why you posted to Applied sciences. Is this content more suitable for the readers of Materials or Dentistry? From the conclusion, it seems that Materials is appropriate.
Answer: The authors thank the reviewer for this suggestion. The authors found that this manuscript is suitable to be included in the Special issue of New Techniques, Materials and Technologies in Dentistry of Applied Sciences journal since it reflects an innovative technique of clasp production using a digital flow of image acquisition, computer-assisted design, and computer-assisted manufacturing by an innovative direct metal laser sintering method.
Q2. Page 2, Line 85-87. The quality of the production largely depends on the experience of the laboratory technician and the quality of the impressions taken by the clinician, which is a time-consuming and expensive method [9,24]. It would be nice to have a concrete number to show how much less error there is in 3D scanning than the sum of traditional errors.
Answer: he authors thank this comment. Comparative studies on RPD production from traditional impression vs. 3D scanning are scarce since digital production of RPD frameworks has yet to evolve enough to justify extensive and reliable studies. Problems in the 3D intra-oral scanning of these cases include difficulty reproducing soft tissue and good images from all dental arch extensions.
Q3. Page 4, Figure 3-5
What is the difference between the acrylic resins in the Figure3-5? Please tell us the characteristics of the three types and the reason for separating the resins. What are the effects of different resins?
Answer: The authors appreciate this comment. The three blocks that supported each of the three teeth during the experiments were fabricated following the same CAD design with 9x20x40 mm transmitted through the same STL file. The material used was light-curing acrylic resins suitable for 3D printing- NextDent Model 2.0, Next Dent, The Netherlands. The name of the resin was added to the manuscript. For less confoundable conclusions, only one figure showing one resin block with tooth number 1 was shown in the manuscript as an example. The rest of the figures were renumbered.
Q4. Page 12, Line 343
These discrepancies may be because the three teeth used for this study have different clinical crown morphology. Is it common to see high reverse action values from 0-4000 cycles in Figure 14? What kind of anatomical difference is due to the fact that only tooth 1 behaves differently?
Answer: The authors appreciate this comment. The retention of a clasp depends not only on the length and width of the retentive arm but on the morphology of the tooth because it can variate the contour of the teeth. The retentive arm should be located above the contour of the teeth, except the tip of the retentive arm, which must be located below the contour. Highly countered tooth usually shows excessive and very extensive retention zone (as tooth number 1), resulting in more retentive arms. Also, variations of the retentive zone can be encountered in the same tooth in the distal or mesial part of the vestibular retentive zone. The fact that the back-action clasp has the tip of the retentive arm located in the part of the vestibular retentive zone and the reverse back-action clasp has the tip in the distal part of the retentive zone can also explain their differences. To decrease the influence of different morphology between and within the teeth, the percentage of retentive forces was calculated in this study and permitted that the variation of retention over time does not depend on the retention of clasps in each tooth in the initial measure.
5) Page 13, Figure 22
The type of microscope is written in the text, and there is little point in including a photograph.
Answer: The authors agreed with this suggestion. Figure 22 was removed from the manuscript, and the rest was renumbered.
6) Page 13, Line 376-381
This part should move to Materials and Methods
Answer: The authors thank this suggestion and moved the paragraph that describes the tooth wear determination to the Materials and Methods chapter.
Reviewer 4 Report
Dear Authority,
The manuscript entitled ‘Evaluation of the retentive forces from removable partial dentures clasps manufactured by the digital method’ investigates variation on retentive forces over time of removable partial denture clasps made of cobalt-chromium SP2 alloy thorough laser sintering process. Clasps designed in two different way; the back-action and the reverse back-action, is involved in insertion and removal test for 20000 cycle. Over 20,000 cycles, a reduced change in retention is observed in the clasps produced by the digital method and suggests few retention over time. For most cycles, there were no differences in the change of retention depending on types of clasps. So that design of clasps has no effect on retention force.
I think, the paper does not include exciting results but the data could be useful for literature for future studies. It could be considered for publication after minor correction according to following comments/recommendations;
1- Some sentences are so long so that it is hard to follow the topic and there are some parts grammatically wrong. Please go over the manuscript again for proof reading
2- Figure 3,4 and 5 could be combined as figure 3 a, b and c. Same issue for Figure 14, 15 and 16.
3- Figure 22 needs to be relocated into Materials and Method section.
The manuscript can be published in Applied Sciences after these minor corrections.
Best wishes,

Some sentences are so long so that it is hard to follow the topic and there are some parts grammatically wrong. Please go over the manuscript again for proof reading
Author Response
Q1: Some sentences are so long so that it is hard to follow the topic and there are some parts grammatically wrong. Please go over the manuscript again for proof reading
Answer: The authors thank the reviewer for this suggestion. Considering the long sentences and grammatical errors, the manuscript was revised and corrected.
Q2: Figure 3,4 and 5 could be combined as figure 3 a, b and c. Same issue for Figure 14, 15 and 16.
Answer: In line with the suggestion of various reviewers, figures 4 and 5 were removed, and figures 14,15 and 16 were combined. The other figures were renumbered.
Q3: Figure 22 needs to be relocated into Materials and Method section.
Answer: In line with the suggestion of various reviewers, figure 22 was removed, and the other figures were renumbered.
Reviewer 5 Report
Manuscript ID applsci-2473194
Journal Applied Sciences (ISSN 2076-3417)
Title
Evaluation of the retentive forces from removable partial dentures clasps manufactured by the digital method
Authors
Vitor Anes * , Cristina B. Neves * , Valeria Bostan , Sérgio Gonçalves , Luís Reis
Dear Author
Do digitally produced clips have a lower impact on the environment?
Please can you cited more references by MDP Journal group ?
Author Response
Q1: Do digitally produced clips have a lower impact on the environment?
Answer: The authors thank this comment. One of the reasons digital technology is essential in this field is to decrease the waste production of the conventional production of removable partial dentures. The acquisition of images, the virtual design of frameworks, and the computer-assisted additive manufacture of prosthetic structures produce less waste and should lower the environmental impact. Nevertheless, subtractive manufacturing has been identified as responsible for more waste than additive manufacturing.
On the other hand, studies should be done on the impact on the environment of these digital techniques since energy consumption could be superior to conventional production in these cases.
Q2: Please can you cited more references by MDP Journal group?
Answer: The authors increase the references chapter with additional MDPI references.